# Complement activation profile of patients with primary focal segmental glomerulosclerosis

**Jing Huang** [1,2,3,4], **Zhao Cui** [1,2,3,4]*, **Qiu-hua Gu** [1,2,3,4], **Yi-miao Zhang** [1,2,3,4], **Zhen Qu** [1,2,3,4], **Xin Wang** [1,2,3,4], **Fang Wang** [1,2,3,4], **Xu-yang Cheng** [1,2,3,4], **Li-qiang Meng** [1,2,3,4], **Gang Liu** [1,2,3,4], **Ming-hui Zhao** [1,2,3,4,5]

**1** Renal Division, Peking University First Hospital, Beijing, PR China, **2** Institute of Nephrology, Peking University, Beijing, PR China, **3** Key Laboratory of Renal Disease, Ministry of Health of China, Beijing, PR China, **4** Key Laboratory of CKD Prevention and Treatment, Ministry of Education of China, Beijing, PR China, **5** Peking-Tsinghua Center for Life Sciences, Beijing, PR China

* cuizhao@bjmu.edu.cn

## Abstract

### Background

Studies on adriamycin mice model suggest complement system is activated and together with IgM contributes to the glomerular injury of primary focal segmental glomerulosclerosis (FSGS). We recently reported primary FSGS patients with IgM and C3 deposition showed unfavorable therapeutic responses and worse renal outcomes. Here we examined the plasma and urinary complement profile of patients with primary FSGS, aiming to investigate the complement participation in FSGS pathogenesis.

### Methods

Seventy patients with biopsy-proven primary FSGS were enrolled. The plasma and urinary levels of C3a, C5a, soluble C5b-9, C4d, C1q, MBL, and Bb were determined by commercial ELISA kits.

### Results

The levels of C3a, C5a and C5b-9 in plasma and urine of FSGS patients were significantly higher than those in normal controls. The plasma and urinary levels of C5b-9 were positively correlated with urinary protein, renal dysfunction and interstitial fibrosis. The plasma C5a levels were positively correlated with the proportion of segmental sclerotic glomeruli. The urinary levels of Bb were elevated, positively correlated with C3a and C5b-9 levels, renal dysfunction, and interstitial fibrosis. The plasma C1q level was significantly decreased, and negatively correlated with urinary protein excretion. Urinary Bb level was a risk factor for no remission (HR = 3.348, 95% CI 1.264–8.870, P = 0.015) and ESRD (HR = 2.323, 95% CI 1.222–4.418, P = 0.010).

**Data Availability Statement:** Data cannot be shared publicly because of our study involved human participants and the ethics committee did not allow us to do so. Data are available from the Peking University First Hospital Clinical Research

Ethics Committee (contact via kyc@bjmu.edu.cn) for researchers who meet the criteria for access to confidential data.

**Funding:** This work was supported by grants from National Natural Science Foundation of China (NSFC) to the Innovation Research Group (81621092) and other NSFC programs (81330020, 81500542, 81622009).

**Competing interests:** The authors have declared that no competing interests exist.

## Conclusion

In conclusion, our results identified the systemic activation of complement in human primary FSGS, possibly via the classical and alternative pathway. The activation of complement system was partly associated with the clinical manifestations, kidney pathological damage, and renal outcomes.

## Introduction

Focal segmental glomerulosclerosis (FSGS) is a group of clinical-pathologic syndromes sharing a common glomerular lesion and mediated by diverse insults directed to or inherent within the podocyte [1–5]. Proteinuria, usually present with nephrotic syndrome, is the most common clinical presentation of primary FSGS. For pathological features, it charactered as focal and segmental glomerular sclerosis by light microscope, and the deposition of immunoglobulin M (IgM) in combination with or without complement 3 (C3) in the sclerotic segment without an obvious granular or linear pattern by immunefluorescence microscopy [6].

In the past few decades, the deposition of IgM and C3 were thought to be nonspecific and made no sense in the pathogenesis of primary FSGS [6]. However, IgM is a pentamer and has stronger ability to activate complement than IgG [7]. Recent studies, using adriamycin-induced glomerulosclerosis animal models, suggested that IgM could activate the complement system within glomeruli, and strategies that reduct of IgM natural antibodies or prevent of complement activation could slow down the progression of glomerulosclerosis [8]. In primary FSGS patients, we recently reported 54.7% of consecutive patients had IgM deposits on the sclerotic segments, and C3 deposits exclusively shown in those patients with IgM glomerular deposit. The patients with IgM and C3 deposition presented unfavorable treatment response and worse renal outcomes [9]. All these findings indicate that IgM deposition may involve disease progression via complement activation.

The activation of complement system could be detected in the circulation and urine of primary FSGS patients [10], which prompts the clinical utility of some complement components as biomarkers for clinical evaluation of kidney injury and outcomes, or as therapeutics targets in FSGS treatments. The pathway of complement activation is still unknown in primary FSGS. In this retrospective study, we measured the plasma and urinary levels of various complement components, reflecting the classical, lectin and alternative pathway of complement activation, in a large cohort of patients with biopsy-proven primary FSGS in active phases. The associations between complement components and clinical-pathologic parameters were analyzed, with the aim to identify the status of complement activation in primary FSGS and their clinical significance.

## Materials and methods

### Patients and samples

Seventy patients with renal biopsy-proven primary FSGS, diagnosed in Peking University First Hospital between 2006 and 2012, were enrolled in this study. They were all treated and followed up for more than two years. Patients with FSGS of known secondary causes, such as virus infection, drug abuse, obesity, or other glomerular diseases including IgA nephropathy, lupus nephritis, pauci-immune glomerulonephritis, and membranous nephropathy, etc., were excluded. Familiar FSGS was excluded by the inquiry of family history of glomerulonephritis.

Complete clinical and pathological data were collected from medical records. Thirty-nine age and gender-matched healthy donors were collected as normal controls.

The research was in compliance of the Declaration of Helsinki and approved by the ethics committee of our hospital before the study began, the full name of our ethics committee is Peking University First Hospital Clinical Research Ethics Committee. Written informed consent was obtained from each participant for tissue, plasma and urine collection, and all data were fully anonymized.

Patients with nephrotic syndrome were treated with corticosteroid combined with immunosuppressive agents including cyclophosphamide, cyclosporine A, tacrolimus, mycophenolate mofetil, leflunomide, et al. Oral prednisone commenced at 1 mg/kg/d for up to 12–16 weeks and then with subsequent tapering, oral cyclophosphamide at 1.5–2 mg/kg/d for 3 months or cyclosporine A at 2–3 mg/kg/d with a trough concentration around 100–150 μg/ml, for 6–12 months. Oral tacrolimus at 0.1–0.2 mg/kg/d with a trough concentration around 5–10 ng/ml. Parts of steroid-resistant patients treated with mycophenolate mofetil and parts of steroid-dependent patients treated with leflunomide at 10mg/d. All patients were treated with angiotensin-converting enzyme inhibitors or angiotensin receptor blockers. For evaluation of therapeutic responses, complete remission was defined as proteinuria less than or equal to 0.3 g/24h with stable serum creatinine (no more than 25% increase from baseline) and serum albumin > 35 g/L. Partial remission was defined as proteinuria less than 3.5 g/24h but greater than 0.3 g/24h, with stable renal function in patients presenting with nephrotic syndrome. Treatment failure was defined as not reaching the criteria of partial remission. Relapse was defined as urinary protein excretion>3.5g/24h after remission. For evaluation of the renal outcomes, the endpoint was end-stage renal disease (ESRD). ESRD was defined as eGFR < 15 ml/min/1.73 m2 or initiation of renal replacement therapy.

## Renal histopathology

Renal biopsy was performed at the time of diagnosis. Renal specimens were evaluated with direct immunofluorescence, light and electron microscopy, and were forwarded to two pathologists. The minimum requirement of glomeruli is 10 glomeruli for FSGS diagnosis by light microscopy, with the median glomeruli of 25.5 (IQR, 18–35) in the current study. Both pathologists examined the biopsies separately, being blinded to each other as well as patients' clinical data. Differences in diagnosis between the two pathologists were resolved by re-reviewing the biopsies and coming to a consensus.

For the seventy patients, the pathological variants were defined according to the Columbia classification [2], which included 23 tip variant, 23 NOS variant, 21 cellular variant, 1 collapsing variant and 2 advanced FSGS (defined as the percentage of sclerosis/total glomeruli over 75%). For direct immunofluorescence, IgG, IgM, IgA, C3c, C1q, fibrinogen and albumin were detected by fluorescein isothiocyanate (FITC)-conjugated antibodies (Dako, Copenhagen, Denmark) on frozen tissues. The fluorescence intensity was determined using a semi-quantitative scale from 0 to 4: 0, negative; 1, weak staining; 2, moderate staining; 3, strong staining; and 4, glaring staining [9]. For light microscopy, paraffin sections were stained with haematoxylin and eosin, periodic acid-schiff, periodic acid-silver methenamine and Masson's trichrome. For electron microscopy, in brief, the tissue was fixed in 2.5% glutaraldehyde and 1% osmium tetroxide, then dehydrated in graded acetone and embedded in Epon 812. Ultrathin sections were cut at a thickness of 80 nm and placed on nickel grids. Then ultrathin sections were stained with uranyl acetate and examined by a transmission electron microscope JEM-1230 (JEOL, Tokyo, Japan).

The renal biopsy findings were categorized according to the Columbia FSGS classification system [2]. Patients with any of the structural manifestations of FSGS were entered into the

registry. Pathologic findings in the glomeruli, tubules, interstitial compartments and vessels were described. The scoring of interstitial fibrosis, interstitial inflammatory cell infiltration, and tubular atrophy was approached by renal pathologists using a modified previously reported system [11–13] as follows: 0, normal; +, <50% of the acreage of interstitium affected; ++, ≥50% of the acreage of interstitium affected in each specimen. Differences in scoring between the pathologists were resolved by re-reviewing the biopsies and reaching a consensus.

## Sample collection

Plasma and first spot urine samples of the patients were collected at the day of renal biopsy. Disodium-EDTA was used as anticoagulant for the plasma samples from patients and controls. The plasma and urine samples were immediately centrifuged at 2000g for 15 min at 4˚C, and then stored in aliquots at -80˚C until use. Repeated freeze/thaw cycles were avoided.

## Quantification of complement components levels

Plasma and urinary levels of human complement components were determined by commercial enzyme-linked immunosorbent assay (ELISA) kits for complement components of soluble C5b-9, C3a, C5a, C4d and Bb (Quidel Corporation, San Diego, CA), and C1q and MBL (Uscn Life Science, Wuhan, China). All the complement components were assayed according to the manufacturer's instructions. The principle of the assay was a four-step procedure: (1) microassay plates were pre-coated with murine monoclonal antibodies binding specifically to the complement components; (2) plasma or urine samples were added according to the optimal dilutions, incubation time and temperature from the instructions; (3) horseradish peroxidase-conjugated antibodies binding to the complement components adsorbed on the plates were added; (4) chromogenic substrate was added to determine the concentration of components. The levels of urinary complement components were corrected by the concentrations of urinary creatinine.

## Statistical analysis

Statistical analysis was performed using statistical software SPSS 16.0 (SPSS Inc., Chicago, IL, USA). For data expression, quantitative data were expressed as mean ± SD (for data that were normally distributed), or median with inter-quartile range (IQR) (for data that were not normally distributed). Differences of quantitative data were assessed using Student's t test (for normally distributed data) or nonparametric test (for two non-normally distributed data or one normally with one non-normally distributed variable). Pearson's correlation test was used to measure the correlation between two normally distributed variables. Spearman's correlation test was used to measure the correlation between two non-normally distributed variables or one normally with one non-normally distributed variable. Univariate survival analysis was carried out using the log-rank test. Multivariate analysis of renal outcomes was performed using the Cox regression model. Multicollinearity was tested using the variance inflation factor (VIF) method among variables. All statistical analyses were two-tailed and $P<0.05$ was considered as significant.

## Results

### Demographic and clinical characteristics of patients with primary FSGS

Seventy patients with renal biopsy-proven primary FSGS, were enrolled. Their demographic, clinical and pathological data are listed in Table 1. The median period between disease onset and sample collection was 3.5 (IQR, 0.65–18) months. Forty-six patients were males and 24

**Table 1. The demographic and clinical parameters of patients with primary FSGS.**

| Parameters | n = 70 |
| --- | --- |
| Gender (male/female) | 46/24 |
| Age (years), (median, IQR) | 28, 19–49 |
| Urine protein (g/24h) (median, IQR) | 7.6, 5.3–13.1 |
| Nephrotic syndrome, n (%) | 65 (92.9%) |
| Serum albumin (g/L) (mean ± s.d.) | 21.1±6.7 |
| Serum creatinine (μmol/l) (median, IQR) | 91.0, 63.5–156.0 |
| eGFR (ml/min/1.73m$^2$) (median, IQR) | 82.7, 42.3–142.2 |
| Renal insufficiency (eGFR<60ml/min/1.73m$^2$), n (%) | 22 (31.4%) |
| Glomeruli with sclerosis (%) (median, IQR) | 8.7, 4.4–20 |
| Pathological variants | |
| tip variant, n (%) | 23 (32.9%) |
| NOS variant, n (%) | 23 (32.9%) |
| cellular variant, n (%) | 21 (30.0%) |
| collapsing variant, n (%) | 1 (1.4%) |
| advanced FSGS, n (%) | 2 (2.8%) |
| Treatment | n = 57 |
| Prednisone, n (%) | 57 (100%) |
| Cyclophosphamide, n (%) | 9 (15.79%) |
| Cyclosporine A, n (%) | 18 (31.58%) |
| Tacrolimus, n (%) | 8 (14.04%) |
| Mycophenolate mofetil, n (%) | 11 (19.30%) |
| Leflunomide, n (%) | 36 (63.16%) |
| Response to treatment | n = 57 |
| Remission, n (%) | 47 (82.4%) |
| Complete remission, n (%) | 41 (71.9%) |
| Partial remission, n (%) | 6 (10.5%) |
| No response, n (%) | 10 (17.5%) |
| Relapse after remission, n (%) | 29/47 (61.7%) |
| Renal dysfunction, n (%) | 8 (14.0%) |
| Follow-up time (months) (median, IQR) | 73.0 (50.4–88.0) |

IQR: inter-quartile range; eGFR: estimated glomerular filtration rate; NOS variant: not otherwise specified variant.

patients were females, with a median age of 28 (13–75) years. The median level of urine protein excretion was 7.6 (5.3–13.1) g/24h and the serum albumin concentration was 21.1±6.7 g/L. Sixty-five (92.9%) patients had nephrotic syndrome. The median serum creatinine level was 91.0 (63.5–156.0) μmol/L and the median estimated glomerular filtration rate (eGFR) was 82.7 (42.3–142.2) ml/min/1.73m$^2$ at the time of renal biopsy. Twenty-two (31.4%) patients presented with decreased eGFR <60 ml/min/1.73m$^2$ at the time of renal biopsy. The pathological variants included 23 tip variants, 23 not otherwise specified (NOS) variants, 21 cellular variants, 1 collapsing variant and 2 advanced FSGS. In 39 healthy donors, 24 were males and 15 were females, with a median age of 28 (15–58) years old.

## Plasma and urinary levels of C3a, C5a and soluble C5b-9

Levels of C3a, C5a and soluble C5b-9 were detected to reflect the activation of complement system. The levels of plasma C3a [(median, IQR) 536.86, 276.59–977.78 vs. 87.40, 55.80–126.31 ng/ml, P<0.001], C5a (9.95, 6.78–16.00 vs. 6.42, 2.68–10.88 ng/ml, P<0.001), and

soluble C5b-9 (381.61, 292.02–516.68 vs. 303.98, 240.24–448.45 ng/ml, P = 0.010) were significantly higher in patients with primary FSGS than that in normal controls (Fig 1).

There were 55/70 patients who had sufficient urinary samples for complement components detection. The clinical and pathological features of these patients were comparable to those of the 70 patients. Urinary levels of complement components were corrected by urinary creatinine (Cr) concentrations. The levels of urinary C3a (8.97, 0.41–67.72 vs. 0.01, 0.00–0.01 ng/mg Cr, P<0.001), C5a (18.69, 1.00–185.91 vs. 0.03, 0.01–0.05 ng/mg Cr, P<0.001), and soluble C5b-9 (59.49, 7.73–216.68 vs. 0.64, 0.07–1.33 ng/mg Cr, P<0.001) were significantly higher in patients with primary FSGS than that in normal controls (Fig 1).

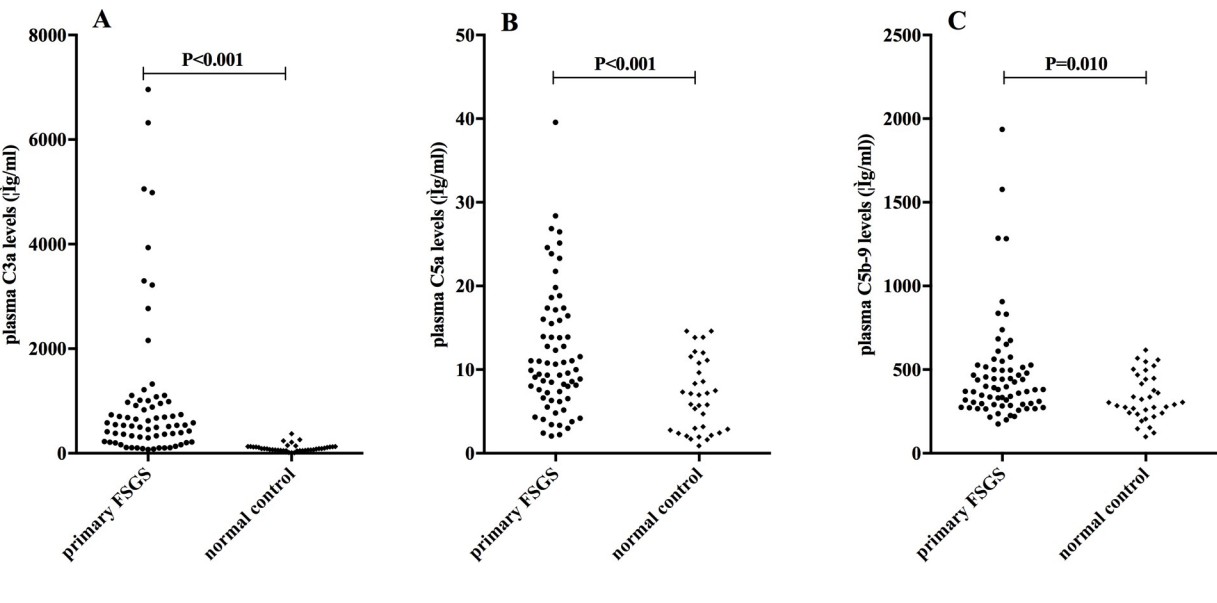

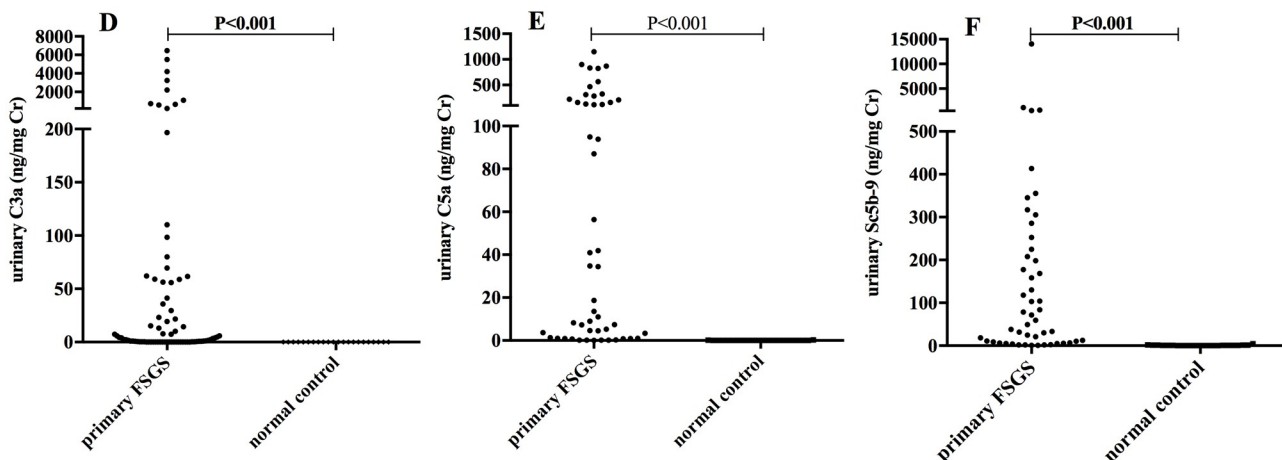

**Fig 1. Plasma and urinary C3a, C5a and soluble C5b-9 levels between patients with primary FSGS and normal subjects.** The plasma and urinary levels of C3a, C5a and C5b-9 were significantly higher in FSGS.

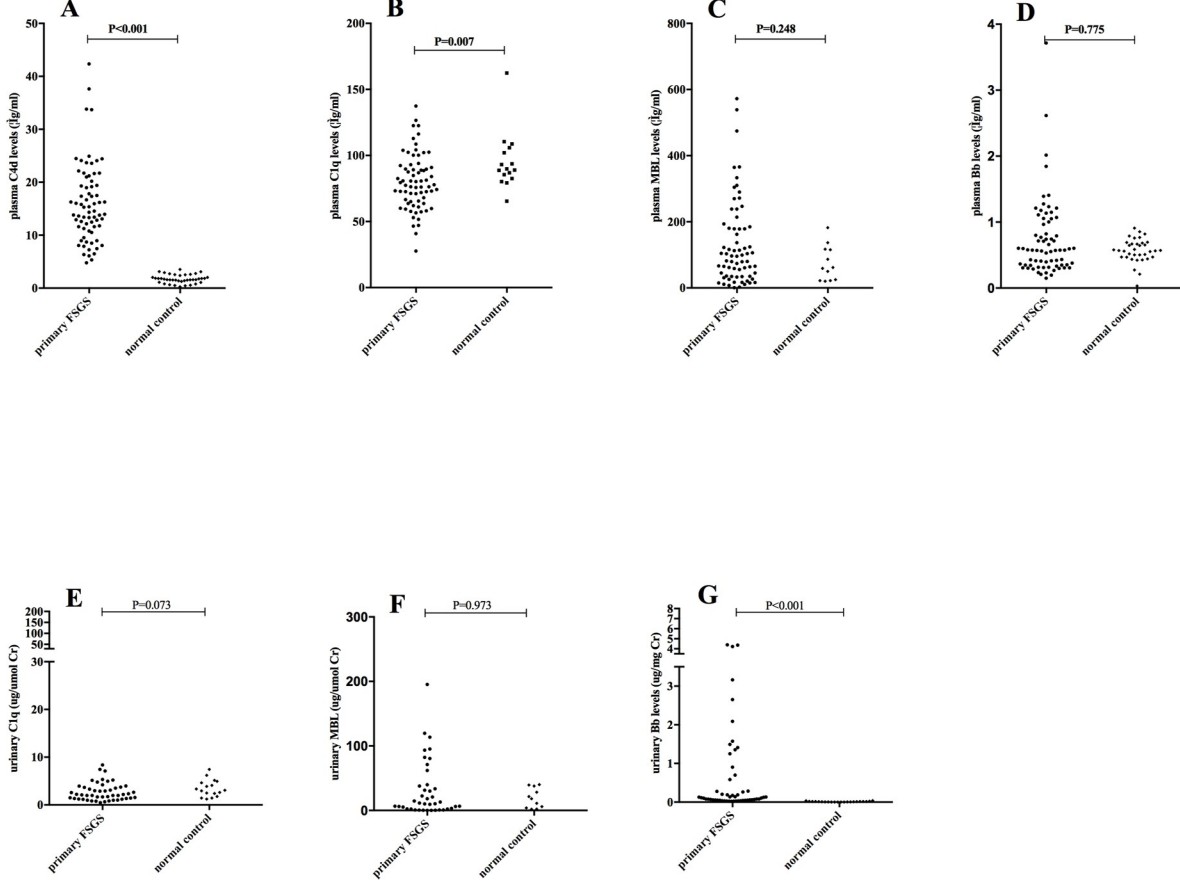

**Fig 2. Plasma and urinary C4d, Bb, C1q and MBL levels between patients with primary FSGS and normal subjects.** The plasma C4d level was significantly increased, the plasma C1q level was significantly decreased, and the urinary Bb was significantly increased in FSGS.

## Plasma and urinary levels of C4d, C1q, MBL, and Bb

Levels of C4d, C1q, MBL, and Bb were examined to explore the pathways of complement activation. The levels of plasma C4d were significantly higher in patients with primary FSGS than that in normal controls (14.55, 11.24–20.19 vs. 1.61, 1.28–2.40 μg/ml, P<0.001). The levels of plasma C1q were significantly lower in patients with primary FSGS than that in normal controls (77.65, 64.77–91.28 vs. 89.38, 83.27–104.93 μg/ml, P = 0.007). The levels of plasma MBL and Bb were comparable between FSGS patients and normal controls (Fig 2).

The levels of urinary Bb were significantly higher in patients with primary FSGS than that in normal controls (0.11, 0.05–0.64 vs. 0.01, 0.00–0.02 μg/mg Cr, P<0.001). The levels of urinary C1q and MBL were comparable between FSGS patients and normal controls (Fig 2).

## Correlations of plasma and urinary complement components

Plasma levels of C4d were positively correlated with that of C3a, C5a and C5b-9 (C3a: r = 0.545, P<0.001; C5a: r = 0.371, P = 0.002; C5b-9: r = 0.430, P<0.001; respectively). Plasma levels of Bb were positively correlated with that of C3a and C5b-9 (C3a: r = 0.447, P<0.001; C5b-9: r = 0.280, P = 0.019; respectively) (Fig 3).

Urinary levels of C1q were positively correlated with that of C3a, C5a and C5b-9 (C3a: r = 0.449, P = 0.001; C5a: r = 0.373, P = 0.006; C5b-9: r = 0.589, P<0.001; respectively). Urinary

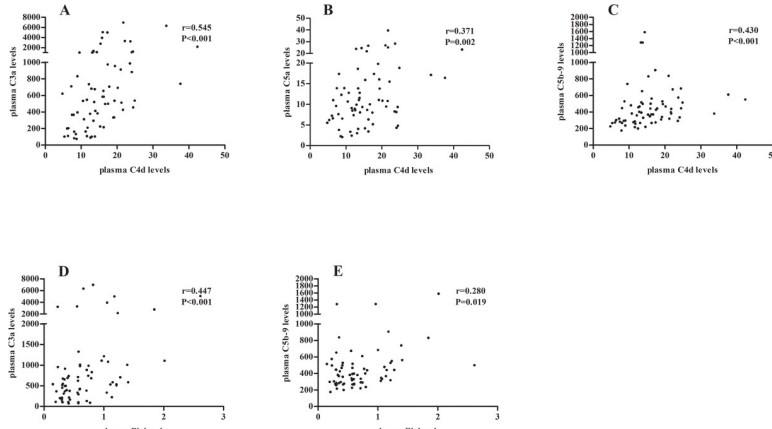

**Fig 3. Correlations of plasma complement components in each patient.** Plasma level of C4d was positively correlated with that of C3a (A), C5a (B) and C5b-9 (C). Plasma level of Bb was positively correlated with that of C3a (D) and C5b-9 (E).

levels of Bb were positively correlated with that of C3a, C5a and C5b-9 (C3a: r = 0.685, P<0.001; C5a: r = 0.569, P<0.001; C5b-9: r = 0.692, P<0.001; respectively) (Fig 4).

For each complement component, the correlation between plasma and urinary levels was analyzed and no correlation was revealed.

## Associations between complement components and clinical-pathologic parameters

Among the 70 patients, the level of plasma C3 was 1.05 ± 0.35 g/L and the level of plasma C4 was 0.27 ± 0.09 g/L. Eleven (15.7%) patients presented with decreased C3 levels, while all patients had normal C4 levels. Compared to those with normal C3, the 11 patients with decreased lower C3 had significantly higher levels of plasma soluble C5b-9 (369.20, 282.37–479.47 vs. 683.4, 370.58–836.44 ng/ml, P = 0.004). Among the 11 patients with lower C3, there were eight patients with cellular variant (72.7%), two with tip variant (18.2%), and one with

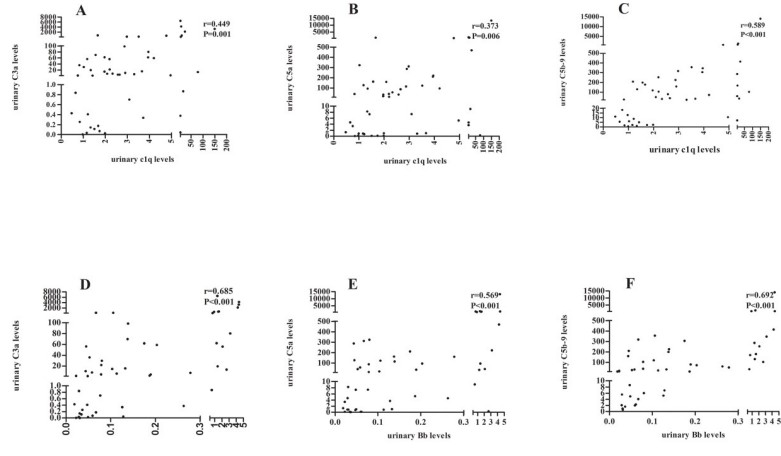

**Fig 4. Correlations of urinary complement components in each patient.** Urinary level of C1q was positively correlated with that of C3a (A), C5a (B) and C5b-9 (C). Urinary level of Bb was positively correlated with that of C3a (D), C5a (E) and C5b-9 (F).

NOS variant (9.1%), which was different from the patients with normal C3, in them there were 13 cellular variant (22.0%), 21 tip variant (35.6%), 22 NOS variant (37.3%), 1 collapsing variant (1.7%) and 2 advanced FSGS (3.4%) (P = 0.021). There was no other difference between these two groups on clinical features, other plasma and urinary complement components, treatment responses or kidney outcomes (P>0.05).

The plasma and urinary levels of soluble C5b-9 were both positively correlated with the urine protein excretion (r = 0.276, P = 0.020; r = 0.455, P = 0.001; respectively). The urinary levels of soluble C5b-9 were further positively correlated with the levels of serum creatinine (r = 0.298, P = 0.037) and negatively correlated with the level of eGFR (r = -0.347, P = 0.014) (Fig 5).

Patients with urinary C3a levels of the upper quartiles had higher urine protein excretion (16.13, 8.73–22.23 vs. 6.42, 4.16–7.75 g/24h, P<0.001), lower levels of serum albumin (17.98 ±3.90 vs. 23.72±6.56 g/L, P = 0.012) and lower levels of eGFR (50.95, 22.64–107.26 vs. 120.65, 80.74–158.40 ml/min/1.73m2, P = 0.005) than patients with urinary C3a levels of the lower quartiles. Patients with urinary C5a levels of the upper quartiles also had higher urine protein excretion (14.20, 6.80–19.41 vs. 6.24, 3.55–7.33 g/24h, P = 0.003), lower levels of serum albumin (17.49±3.35 vs. 23.71±7.40 g/L, P = 0.012) and lower eGFR (50.95, 22.64–68.42 vs. 132.96, 90.37–158.40 ml/min/1.73m2, P<0.001) than patients with urinary C5a levels of the lower quartiles.

The plasma levels of C1q were negatively correlated with the urine protein excretion (r = -0.370, P = 0.002), and positively correlated with the levels of serum albumin (r = 0.392,

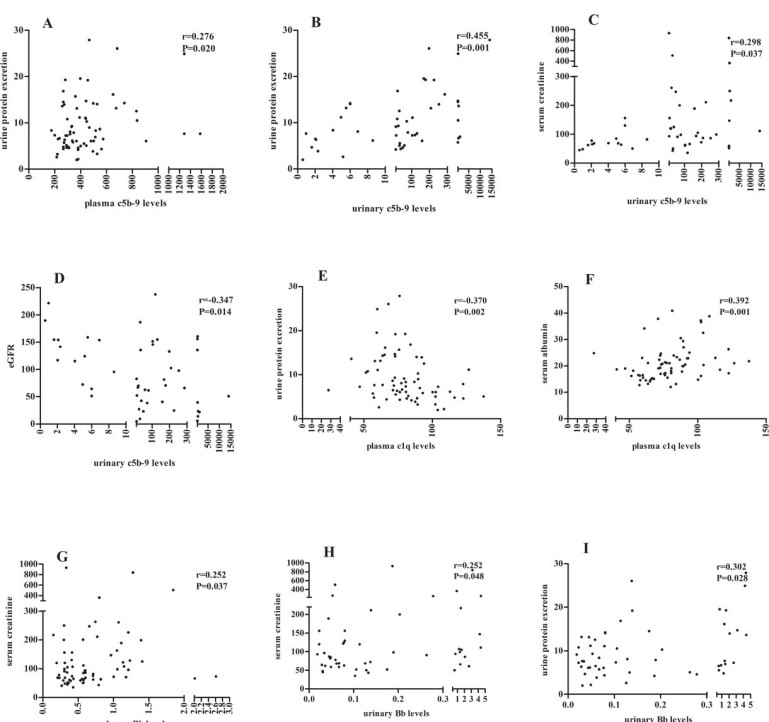

**Fig 5. Associations between complement components and clinical-pathologic parameters.** The plasma (A) and urinary (B) levels of soluble C5b-9 were positively correlated with urinary protein excretion. The urinary C5b-9 level was positively correlated with the concentration of serum creatinine (C) and negatively correlated with eGFR (D). The plasma level of C1q was negatively correlated with urinary protein excretion (E) and positively correlated with serum albumin (F). The plasma (G) and urinary (H) level of Bb was positively correlated with serum creatinine. The urinary Bb level was positively correlated with urinary protein (I).

**Table 2. Plasma levels of complement components in different pathological variants of primary FSGS patients.**

| Pathological variant | Plasma levels of complement components (median, IQR) | | | | | | |
|---|---|---|---|---|---|---|---|
| | C1q (µg/ml) | MBL (ng/ml) | Bb (µg/ml) | C4d (µg/ml) | C3a (ng/ml) | C5a (ng/ml) | C5b-9 (ng/ml) |
| Tip variants | 76.31 | 106.16 | 0.57 | 13.82 | 512.12 | 9.60 | 392.64 |
| (n = 23) | (71.35–88.87) | (55.70–217.56) | (0.39–0.77) | (10.51–20.19) | (133.01–954.44) | (7.00–11.68) | (318.21–497.38) |
| Nos variants | 84.10 | 96.36 | 0.57 | 16.27 | 497.85 | 9.11 | 336.13 |
| (n = 23) | (73.24–104.22) | (34.02–180.32) | (0.37–0.79) | (11.61–21.74) | (357.27–1062.41) | (6.18–19.81) | (267.21–180.32) |
| Cellular variants | 66.80 | 99.49 | 0.55 | 13.78 | 654.57 | 13.90 | 513.92 |
| (n = 21) | (58.49–8369) | (40.50–170.25) | (0.31–1.18) | (10.16–18.55) | (339.52–1056.99) | (9.00–17.99) | (376.10–784.76) |
| P value | | | | | | | |
| Total | 0.022 | 0.548 | 0.995 | 0.699 | 0.643 | 0.132 | 0.003 |
| tip vs. NOS | 0.132 | | | | | | 0.047 |
| tip vs. cellular | 0.181 | | | | | | 0.076 |
| NOS vs. cellular | 0.006 | | | | | | 0.001 |

IQR: inter-quartile range; NOS: not otherwise specified.

$P = 0.001$). The plasma levels ($r = 0.252$, $P = 0.037$) and urinary levels ($r = 0.252$, $P = 0.048$) of Bb were positively correlated with the levels of serum creatinine. The urinary levels of Bb were also positively correlated with the levels of urine protein ($r = 0.302$, $P = 0.028$) (Fig 5).

The plasma levels of C5a were positively correlated with the number of segmental sclerosis glomeruli ($r = 0.320$, $P = 0.022$). Compared to the tip and NOS variant, patients with cellular variant had the highest level of plasma soluble C5b-9, the lowest level of plasma C1q, and the highest levels of urinary C1q, Bb, C3a, C5a, and soluble C5b-9 (Table 2 and Table 3).

The patients with interstitial fibrosis over 50% of renal biopsy sections had higher urinary levels of soluble C5b-9 (578.23, 55.04–7357.78 vs. 35.72, 5.99–193.16 µg/mg, $P = 0.025$), higher plasma Bb (0.08, 0.64–1.17 vs. 0.53, 0.32–0.75 µg/ml, $P = 0.008$) and urinary Bb (0.55, 0.12–4.37 vs. 0.08, 0.05–0.28 µg/mg, $P = 0.040$) levels, and higher urinary C1q (4.22, 2.96–43.64 vs. 1.99, 1.21–3.94 ng/µmol, $P = 0.021$) levels.

**Table 3. Urinary levels of complement components in different pathological variants of primary FSGS patients.**

| Pathological variant | Urinary levels of complement components (median, IQR) | | | | | |
|---|---|---|---|---|---|---|
| | C1q | MBL | Bb | C3a | C5a | C5b-9 |
| | (µg /mg Cr) | (ng/mg Cr) | (µg/mg Cr) | (ng/mg Cr) | (ng/mg Cr) | (ng/mg Cr) |
| Tip variants | 1.42 | 8.38 | 0.05 | 1.31 | 3.67 | 8.91 |
| (n = 23) | (0.84–2.87) | (1.14–52.05) | (0.03–0.08) | (0.10–31.20) | (0.16–286.59) | (4.28–58.10) |
| Nos variants | 2.20 | 5.25 | 0.13 | 4.33 | 4.56 | 37.94 |
| (n = 23) | (1.97–3.73) | (0.68–20.21) | (0.07–0.29) | (0.34–15.32) | (0.85–34.78) | (6.88–103.47) |
| Cellular variants | 3.74 | 59.22 | 0.20 | 80.02 | 160.96 | 208.15 |
| (n = 21) | (1.51–5.95) | (15.57–108.95) | (0.06–1.58) | (19.31–721.32) | (60.92–824.25) | (30.04–413.69) |
| P value | | | | | | |
| Totle | 0.020 | 0.002 | 0.009 | <0.001 | <0.001 | 0.005 |
| tip vs. NOS | 0.019 | 0.347 | 0.014 | 0.957 | 0.681 | 0.186 |
| tip vs. cellular | 0.015 | 0.008 | 0.006 | 0.001 | 0.005 | 0.005 |
| NOS vs. cellular | 0.391 | 0.001 | 0.311 | <0.001 | <0.001 | 0.012 |

IQR: inter-quartile range; NOS: not otherwise specified.

Table 4. Univariate and multivariate Cox regression analysis for the renal outcomes of primary FSGS patients.

| | Univariate Cox analysis | | | Multivariate Cox analysis | | |
|---|---|---|---|---|---|---|
| | HR | 95% CI | *P*-value | HR | 95% CI | *P*-value |
| Gender | 0.256 | 0.061–1.074 | 0.062 | | | |
| Age | 0.987 | 0.947–1.028 | 0.524 | | | |
| Serum creatinine | 1.003 | 1.000–1.006 | 0.040 | | | |
| Urinary Bb level | 2.548 | 1.437–4.519 | 0.001 | 2.323 | 1.222–4.418 | 0.010 |
| Urinary C3a level | 1.001 | 1.000–1.002 | 0.004 | | | |
| Urinary C5a level | 1.003 | 1.001–1.006 | 0.005 | | | |
| C3 glomerular deposition | 10.957 | 2.603–46.132 | 0.001 | 10.036 | 1.328–75.856 | 0.025 |
| Interstitial fibrosis | 13.278 | 3.111–56.674 | <0.001 | | | |

HR: hazard ratio; CI: confidence intervals

## Association between the complement levels and treatment responses and renal outcomes

Fifty-seven of the 70 primary FSGS patients were followed up with a median time of 73.0 (50.4–88.0) months. Among them, 47 (82.4%) patients achieved remission, including 41 (71.9%) patients with complete remission and 6 (10.5%) patients with partial remission. Patients who did not achieve remission had higher levels of urinary Bb (1.41, 0.08–3.16 vs. 0.08, 0.04–0.19 μg/mg Cr, P = 0.012), C3a (80.02, 4.04–721.32 vs. 4.33, 0.21–46.01 ng/mg Cr, P = 0.027), C5a (220.89, 34.78–824.25 vs. 7.31, 0.86–104.72 ng/mg Cr, P = 0.001) and C5b-9 (247.51, 117.11–419.68 vs. 27.34, 5.44–107.44 ng/mg Cr, P = 0.004). Using binary logistic regression, we found that the higher level of urinary Bb was a risk factor for no remission [HR (hazard ratio) = 3.348, 95% CI (confidence interval) 1.264–8.870, P = 0.015].

In the 47 remission patients, 29 (61.7%) patients underwent relapse during follow-up. The plasma and urinary complement levels were not associated with disease relapse.

During follow up, 8/57 (14.0%) patients went into end stage renal disease (ESRD). Using univariate survival analysis, we found that the levels of urinary Bb (HR = 2.548, 95%CI 1.437–4.519, P = 0.001), C3a (HR = 1.001, 95%CI 1.000–1.002, P = 0.004), C5a (HR = 1.003, 95%CI 1.001–1.006, P = 0.005) were risk factors for ESRD. Other risk factors included baseline Scr (HR = 1.003, 95%CI 1.000–1.006, P = 0.040), C3 glomerular deposition (HR = 10.957, 95%CI 2.603–46.132, P = 0.001) and interstitial fibrosis ≥ 50% (HR = 13.278, 95%CI 3.111–56.674, P<0.001). Using multivariate Cox analysis, we found that the level of urinary Bb (HR = 2.323, 95% CI 1.222–4.418, P = 0.010) and C3 glomerular deposition (HR = 10.036, 95% CI 1.328–75.856, P = 0.025) were independent risk factors for ESRD (Table 4). The multicollinearity was tested by calculating variance inflation factor (VIF) between variables and no obvious collinearity was found (all VIFs < 5).

## Discussion

Complement activation has been identified in adriamycin-induced animal models of glomerulosclerosis [14, 15], which is associated with the progression of glomerulosclerosis. Glomerular IgM and C3 deposition were also observed in 54.7% of primary FSGS patients on the sclerotic segments and were associated with therapeutic responses and renal outcomes [9]. In the current study, we measured the plasma and urinary levels of complement components in primary FSGS patients, and identified that the complement system is activated through the alternative pathway and possiblely classical pathway, and participates in the kidney damage by membrane attack complex and C5a, and may affect the kidney prognosis.

We found that the plasma and urinary levels of C3a, C5a and soluble C5b-9 in patients with primary FSGS were all significantly increased, which indicated that the complement system was activated in the circulation of patients with primary FSGS. Turnberg et al. have reported that the C3 deficient adriamycin nephropathy mice showed a delay of the onset of detectable albuminuria, a reduction of foot-processes effacement, thus a less glomerulosclerosis and less tubulointersititial scarring, and better renal function for a long-time observation [14]. Thurman et al. have also reported that the levels of C5b-9 in plasma from patients with FSGS were significantly higher than that in healthy controls [10]. These findings demonstrate complement system activation in primary FSGS, both in human and in animal models, and indicate the participation of complement activation in renal damage.

Once activated, the complement system may mediate renal damage through either membrane attack complex (MAC) or via the leukocyte chemoattractant effect of C5a. We found that both the circulating and urinary levels of soluble C5b-9 were positively correlated with the urinary protein excretion of FSGS patients. The urinary levels of C5b-9 were further correlated with measures of renal function. Patients with severe interstitial fibrosis on renal biopsy often presented with higher levels of urinary C5b-9. These results revealed that the renal damage might be mediated by MAC in human FSGS. In mCd59a-/- mice, which lack the major regulator of C5b-9 formation, adriamycin nephropathy develops significantly exacerbated renal disease in terms of more glomerulosclerosis and tubulointerstitial injury [14]. In C6-deficient rats, which are unable to form MAC, it has been shown that adriamycin nephropathy were protected from peritubular myofibroblast accumulation and interstitial extracellular matrix deposition when they had equivalent degree of proteinuria with wild-type controls [16]. These results demonstrate that the activation of complement cascade leading to the MAC generation is a principal mediator of glomerulosclerosis and tubulointerstitial injury in FSGS.

In the present study, we firstly found that the circulating and urinary C5a levels were significantly elevated in human FSGS patients. The plasma levels of C5a were positively correlated with the prevalence of segmental sclerosis in glomeruli, urinary C5a levels were correlated with renal outcomes, which indicate the potential mechanism of glomerular damage mediated through C5a. Anaphylatoxin C5a targets a broad spectrum of immune and non-immune cells to induce inflammatory response, regulates vasodilation, increases the permeability of small blood vessels, mediates oxidative burst, and promote fibrogenesis, et al [17–20]. There are few literatures on C5a in both human and animal FSGS. Conventional immunosuppressive treatments, such as corticosteroids and calcineurin inhibitors, have limited efficacy in reducing complement activation [21]. Newer drugs, that target complement system, have been attempted in some diseases, such as atypical hemolytic uremic syndrome, ANCA associated vasculitis, C3 glomerulopathy, etc [22, 23]. Our findings raise the possibility to identify the patients likely to benefit from complement treatment like eculizumab [24].

The pathways of complement activation in primary FSGS patients remain unclear. Thurman et al. reported plasma C4a, Bb levels was significantly higher in the FSGS patients compared to healthy controls, but they did not test MBL levels in plasma or urine. In the current study, we found that the urinary levels of Bb were significantly increased, which was also reported by Morita et al. [25]. We firstly revealed that both the circulating and urinary levels of Bb were positively correlated with the levels of C3a and C5b-9. The plasma and urinary Bb levels were both associated with the clinical-pathologic features in terms of renal function and interstitial fibrosis, and urinary Bb level was a risk factor for therapeutic response and renal outcome. These results demonstrate that the alternative pathway of complement was activated and play an important role in the pathogenesis of human primary FSGS. Using the adriamycin-induced renal injury resembling FSGS, Lenderink et al. [15] found that complement activation was attenuated in the glomeruli and tubulointerstitium of fB-/- mice compared with

wild-type controls, demonstrating that complement activation does occur through the alternative pathway.

IgM deposit is usually observed on the sclerotic lesion of FSGS patients. Strassheim et al. [8] found that IgM bound to injury associated glomerular epitopes and then increased classical pathway activation in mice with adriamycin-induced glomerulosclerosis. Panzer et al. [26] found that IgM exacerbates glomerular disease progression in complement-induced glomerulopathy. However, the C1q-deficient mice developed similar disease to wild-type controls, after adriamycin treatment, which suggests the lack of classical complement activation in FSGS animal models [15]. In the current study, we found the decreased level of C1q and increased levels of C4d in the circulation of FSGS patients. Plasma levels of C1q were negatively associated with urine protein excretion, and urinary C1q levels were positively associated with severe interstitial fibrosis. Our results indicate the involvement of classic complement activation in primary FSGS, which need further investigations.

For each complement component, we found no correlation between its plasma levels and urinary levels, which argues other sources of urinary complement besides circulating filtration from basement membrane. Different pathological variants of FSGS showed different levels of complement activation, with more active in cellular variants than tip and NOS variants. In the current study, we found the patients with higher levels of urinary C3a and C5a have more severe clinical manifestation and worse renal function, but the plasma C3a and C5a did not show these differences. We also found the association of therapeutic responses and renal outcomes to the urinary levels of Bb, C3a, C5a and C5b-9, but not to the plasma complement levels. So we suggest that the activation of complement in the kidney in-situ also take part in the pathogenesis of primary FSGS, which needs further studies on kidney tissues and animal experiments.

## Conclusions

In conclusion, we provide evidence for the systemic activation of complement in human primary FSGS, via the alternative pathway and possibly classical pathway. Kidney damage was mediated by MAC and C5a, and renal outcome was associated with urinary Bb level. These findings value the complement activation in the mechanism of primary FSGS and bring possible consideration of complement therapies in treatment strategies. But there were parts of patients have follow-up data in our study, RCTs of large cohort and long term of follow-up are needed to verify the results.

## Acknowledgments

The technical support by Lu Bai was greatly appreciated.

## Author Contributions

**Conceptualization:** Jing Huang, Zhao Cui, Gang Liu, Ming-hui Zhao.

**Data curation:** Jing Huang, Qiu-hua Gu, Zhen Qu.

**Formal analysis:** Jing Huang, Qiu-hua Gu, Yi-miao Zhang, Xin Wang, Fang Wang, Xu-yang Cheng, Li-qiang Meng, Gang Liu.

**Resources:** Yi-miao Zhang, Zhen Qu, Xin Wang, Fang Wang, Xu-yang Cheng, Li-qiang Meng.

**Writing – original draft:** Jing Huang.

**Writing – review & editing:** Jing Huang, Zhao Cui, Gang Liu, Ming-hui Zhao.

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
