## [Decision Letter · Decision Letter 0]

3 Apr 2020

PONE-D-20-06891

Complement activation profile of patients with primary focal segmental glomerulosclerosis

PLOS ONE

Dear Dr. huang,

Thank you for submitting your manuscript to PLOS ONE. After careful consideration, we feel that it has merit but does not fully meet PLOS ONE’s publication criteria as it currently stands. Therefore, we invite you to submit a revised version of the manuscript that addresses the points raised during the review process.

We would appreciate receiving your revised manuscript by May 18 2020 11:59PM. To enhance the reproducibility of your results, we recommend that if applicable you deposit your laboratory protocols in protocols.io, where a protocol can be assigned its own identifier (DOI) such that it can be cited independently in the future. For instructions see: http://journals.plos.org/plosone/s/submission-guidelines#loc-laboratory-protocols

We look forward to receiving your revised manuscript.

Kind regards,

Zhanjun Jia

Academic Editor

PLOS ONE

Additional Editor Comments (if provided):

Please address the concerns from the reviewers.

Journal Requirements:

2. In the ethics statement in the manuscript and in the online submission form, please provide additional information about the patient records used in your retrospective study. Specifically, please ensure that you have discussed whether all data were fully anonymized before you accessed them and/or whether the IRB or ethics committee waived the requirement for informed consent. If patients provided informed written consent to have data from their medical records used in research, please include this information.

Please state in the manuscript text that this is a retrospective study.

3.Thank you for including your ethics statement:  "The research was in compliance of the Declaration of Helsinki and approved by the ethics committee of our hospital.".   

For studies reporting research involving human participants, PLOS ONE requires authors to confirm that this specific study was reviewed and approved by an institutional review board (ethics committee) before the study began.

4. Please include your tables as part of your main manuscript and remove the individual files. Please note that supplementary tables (should remain/ be uploaded) as separate "supporting information" files

"This work was supported by grants from National Natural Science Foundation of China (NSFC) to the Innovation

Research Group (81621092) and other NSFC programs (81330020, 81500542, 81622009)."

7. PLOS requires an ORCID iD for the corresponding author in Editorial Manager on papers submitted after December 6th, 2016. Please ensure that you have an ORCID iD and that it is validated in Editorial Manager. To do this, go to ‘Update my Information’ (in the upper left-hand corner of the main menu), and click on the Fetch/Validate link next to the ORCID field. This will take you to the ORCID site and allow you to create a new iD or authenticate a pre-existing iD in Editorial Manager. Please see the following video for instructions on linking an ORCID iD to your Editorial Manager account: https://www.youtube.com/watch?v=_xcclfuvtxQ

Reviewers' comments:

Reviewer's Responses to Questions

**Comments to the Author**

1. Is the manuscript technically sound, and do the data support the conclusions?

Reviewer #1: Yes

Reviewer #2: Partly

2. Has the statistical analysis been performed appropriately and rigorously? 

Reviewer #1: Yes

Reviewer #2: Yes

3. Have the authors made all data underlying the findings in their manuscript fully available?

Reviewer #1: Yes

Reviewer #2: Yes

4. Is the manuscript presented in an intelligible fashion and written in standard English?

Reviewer #1: No

Reviewer #2: No

5. Review Comments to the Author

Reviewer #1: The article ‘Complement activation profile of patients with primary focal segmental glomerulosclerosis’ written by Dr. Huang Jing illustrated that activation of complement system was positively correlated with more severe clinical manifestations, compromised kidney function and poor renal outcomes. There exists systemic activation of complement in human primary FSGS, possibly via alternative and classical pathways. The large cohort of patients, proper study design and more than 2 years follow-up make the study sound and reliable. The study paved the way for possible complement monoclonal antibody used in FSGS. However, there’s still some minor revisions for the article.

Minor revisions

1. An English revision, including grammars and spelling, like para 1 line 3: ’presents with’, para 1 line 5’characters’, para 2 line 6 ‘reduction…prevention’, para 2 line 9 ’having’ and so on, is mandatory.

2. Please show the demographic information of the gender and age-matched healthy control group.

3. As shown in table 1, serum albumin level varied from 14.4g/L to 27.8g/L and 24h protein excretions were within massive range. Therefore, complement components could also leaked out into the urine. Is it necessary to correct plasma complement level with serum albumin level?

Reviewer #2: This is a nice study to evaluate the plasma and urinary complement profile of patients with primary FSGS. But the results section of the manuscript is too complicated and not easy to follow. I would be easy read if the authors could simplify results section. In addition, for the conclusion of "The activation of complement system was associated with the clinical manifestations, kidney pathological damage, and renal outcomes", I do not find enough data to support the activation of complement system was associated with the clinical manifestations and kidney pathological damage. I have following comments which should be considered.

1. The definition of complete remission was stricter than other studies which was that proteinuria ≤ 0.3 g/24h with stable serum creatinine and serum albumin great than 35 g/L. Why the authors define complete remission as proteinuria less than or equal to 0.15 g/24h with stable serum creatinine?

2. The endpoint of renal outcome is end stage of renal disease, please clarify the definition.

3. I would suggest move the sentence of "The pathological variants of the 70 patients were defined according to the Columbia classification [2], which included 23 tip variant, 23 NOS variant, 21 cellular variant, 1 collapsing variant and 2 advanced FSGS (defined as the percentage of sclerosis/total glomeruli over 75%)" to renal histopathology section to make easy understanding.

4. A total of 70 patients were included in this study, but in Table 1 only 57 patients were evaluated the response to treatment, please clarify the reason of not evaluation 70 patients. The authors mentioned treatment in the method section, but did not include them in Table 1. It would helpful to include them in Table 1.

5. For Cox regression, did the authors included urinary Bb, C3a and C5a in the same model? Urinary Bb was correlated with C3a and C5a, therefore, the issue of collinearity will produce when all of them were included in the same model.

6. PLOS authors have the option to publish the peer review history of their article (what does this mean?). If published, this will include your full peer review and any attached files.

Reviewer #1: Yes: Ruochen Che

Reviewer #2: No

---

## [Author Response · Author response to Decision Letter 0]

25 May 2020

Dear Editor,

Re: PONE-D-20-06891, entitled “Complement activation profile of patients with primary focal segmental glomerulosclerosis”

Thank you for giving us the opportunity to revise our above referenced manuscript. We have revised the manuscript carefully according to the Editor and Reviewers’ comments and recommendations. The removed contents were highlighted in blue line-through and the revisions were highlighted in red line in the text according to your requirement. 

In brief, the revision includes the following changes:

1. We have stated this is a retrospective study in the revised manuscript.

2. We added the name of our ethics committee. 

3. The manuscript have been revised according to your style requirements. 

4. The demographic information of healthy controls have been added.

5. The treatment of primary FSGS patients have been added in table 1.

6. The multicollinearity was tested among urinary Bb, C3a and C5a.

The response to each question or comment of the reviewers is listed point-by-point in the attached pages. We hope the revised form is suitable for publication in your journal.

Yours sincerely, 

Jing Huang, Zhao Cui, Qiu-hua Gu, Yi-miao Zhang, Zhen Qu, Xin Wang, Fang Wang, Xu-yang Cheng, Li-qiang Meng, Gang Liu, Ming-hui Zhao.

Editor Comments:

1. Please ensure that your manuscript meets PLOS ONE's style requirements, including those for file naming. The PLOS ONE style templates can be found at http://www.plosone.org/attachments/PLOSOne_formatting_sample_main_body.pdf andhttp://www.plosone.org/attachments/PLOSOne_formatting_sample_title_authors_affiliations.pdf

Answer: Thank you. We have revised the manuscript according to your style requirements.

2. In the ethics statement in the manuscript and in the online submission form, please provide additional information about the patient records used in your retrospective study. Specifically, please ensure that you have discussed whether all data were fully anonymized before you accessed them and/or whether the IRB or ethics committee waived the requirement for informed consent. If patients provided informed written consent to have data from their medical records used in research, please include this information. Please state in the manuscript text that this is a retrospective study.

Answer: Thank you. We ensure that all data were fully anonymized and all patients had provided informed written consent to have data from their medical records used in research. These were added in the revised manuscript (page 6, paragraph 2). We also have stated that this is a retrospective study in the revised manuscript text (page 5, paragraph 1, line 5).

3.Thank you for including your ethics statement: "The research was in compliance of the Declaration of Helsinki and approved by the ethics committee of our hospital.". 

For studies reporting research involving human participants, PLOS ONE requires authors to confirm that this specific study was reviewed and approved by an institutional review board (ethics committee) before the study began. Please amend your current ethics statement to include the full name of the ethics committee/institutional review board(s) that approved your specific study. Once you have amended this/these statement(s) in the Methods section of the manuscript, please add the same text to the “Ethics Statement” field of the submission form (via “Edit Submission”).

Answer: We confirm that our study was reviewed and approved by the ethics committee of our hospital before the study began. The full name of our ethics committee is Peking University First Hospital Clinical Research Ethics Committee, we have amended in both methods section of the manuscript (page 6, paragraph 2, line 2-3) and ‘Ethics Statemen’ of the submission form.

4. Please include your tables as part of your main manuscript and remove the individual files. Please note that supplementary tables (should remain/ be uploaded) as separate "supporting information" files.

Answer: I’m sorry for the mistake. We have removed the individual table files and revised tables as part of the main manuscript (page 12-13, page 19, page 20, page 23).

5. Thank you for stating the following in the Acknowledgments Section of your manuscript:"This work was supported by grants from National Natural Science Foundation of China (NSFC) to the Innovation Research Group (81621092) and other NSFC programs (81330020, 81500542, 81622009)." We note that you have provided funding information that is not currently declared in your Funding Statement. However, funding information should not appear in the Acknowledgments section or other areas of your manuscript. We will only publish funding information present in the Funding Statement section of the online submission form. Please remove any funding-related text from the manuscript and let us know how you would like to update your Funding Statement. Currently, your Funding Statement reads as follows: The author(s) received no specific funding for this work.

Answer: Thank you. We have removed funding-related text from the manuscript (page 28, paragraph 3, line 1-4) and updated our Funding Statement in the section of the online submission form. We also added our Funding Statement in the revised cover letter.

6. In your Data Availability statement, you have not specified where the minimal data set underlying the results described in your manuscript can be found. PLOS defines a study's minimal data set as the underlying data used to reach the conclusions drawn in the manuscript and any additional data required to replicate the reported study findings in their entirety. All PLOS journals require that the minimal data set be made fully available. For more information about our data policy, please see http://journals.plos.org/plosone/s/data-availability.Upon re-submitting your revised manuscript, please upload your study’s minimal underlying data set as either Supporting Information files or to a stable, public repository and include the relevant URLs, DOIs, or accession numbers within your revised cover letter. For a list of acceptable repositories, please see http://journals.plos.org/plosone/s/data-availability#loc-recommended-repositories. Any potentially identifying patient information must be fully anonymized. Important: If there are ethical or legal restrictions to sharing your data publicly, please explain these restrictions in detail. Please see our guidelines for more information on what we consider unacceptable restrictions to publicly sharing data: http://journals.plos.org/plosone/s/data-availability#loc-unacceptable-data-access-restrictions. Note that it is not acceptable for the authors to be the sole named individuals responsible for ensuring data access. We will update your Data Availability statement to reflect the information you provide in your cover letter.

Answer: I’m sorry that we could not upload our study’s minimal underlying data set and share our data publicly, because our study involved human participants and the ethics committee did not allow us to do so. All of the authors know this. Data are available from the Peking University First Hospital Clinical Research Ethics Committee (contact via kyc@bjmu.edu.cn) for researchers who meet the criteria for access to confidential data. It has been added in the revised cover letter.

7. PLOS requires an ORCID iD for the corresponding author in Editorial Manager on papers submitted after December 6th, 2016. Please ensure that you have an ORCID iD and that it is validated in Editorial Manager. To do this, go to ‘Update my Information’ (in the upper left-hand corner of the main menu), and click on the Fetch/Validate link next to the ORCID field. This will take you to the ORCID site and allow you to create a new iD or authenticate a pre-existing iD in Editorial Manager. 

Answer: Thank you. I have updated my information and got an ORCID ID as 0000-0002-1755-5008. The ORCID ID for Zhao Cui is 0000-0002-5837-1926.

Reviewers' comments:

Reviewer's Responses to Questions

Comments to the Author

1. Is the manuscript technically sound, and do the data support the conclusions?

Reviewer #1: Yes

Reviewer #2: Partly

2. Has the statistical analysis been performed appropriately and rigorously?

Reviewer #1: Yes

Reviewer #2: Yes

3. Have the authors made all data underlying the findings in their manuscript fully available?

Reviewer #1: Yes

Reviewer #2: Yes

4. Is the manuscript presented in an intelligible fashion and written in standard English?

Reviewer #1: No

Reviewer #2: No

5. Review Comments to the Author

Reviewer #1: 

The article ‘Complement activation profile of patients with primary focal segmental glomerulosclerosis’ written by Dr. Huang Jing illustrated that activation of complement system was positively correlated with more severe clinical manifestations, compromised kidney function and poor renal outcomes. There exists systemic activation of complement in human primary FSGS, possibly via alternative and classical pathways. The large cohort of patients, proper study design and more than 2 years follow-up make the study sound and reliable. The study paved the way for possible complement monoclonal antibody used in FSGS. However, there’s still some minor revisions for the article.

Minor revisions

1. An English revision, including grammars and spelling, like para 1 line 3: ’presents with’, para 1 line 5’characters’, para 2 line 6 ‘reduction…prevention’, para 2 line 9 ’having’ and so on, is mandatory.

Answer: Thank you for your comments. We have revised our manuscript according to your suggestions. The English writing was reviewed by an editor of an English journal.

2. Please show the demographic information of the gender and age-matched healthy control group.

Answer: Thank you. In 39 healthy donors, 24 were males and 15 were females, with a median age of 28 (15-58) years old. It has been added in the revised manuscript (page 10, paragraph 2, line 12; page 11, paragraph 1, line 1).

3. As shown in table 1, serum albumin level varied from 14.4g/L to 27.8g/L and 24h protein excretions were within massive range. Therefore, complement components could also leaked out into the urine. Is it necessary to correct plasma complement level with serum albumin level? 

Answer: Thank you very much. We are sorry for our neglect, we did not test serum albumin levels in normal controls when we test complement profiles. But in our study, the results showed plasm C3a, C5a, sC5b-9, C4d were significantly higher in patients with primary FSGS than that in normal controls, but plasma C1q were significantly lower, and plasma MBL and Bb were comparable, not all complement levels in FSGS patients were lower than those in normal controls, so we think it would not Influence the results significantly. And we have some difficulties in testing serum albumin levels in normal controls now, so we think it might acceptable did not correct plasma complement level with serum albumin level.

Reviewer #2: 

This is a nice study to evaluate the plasma and urinary complement profile of patients with primary FSGS. But the results section of the manuscript is too complicated and not easy to follow. I would be easy read if the authors could simplify results section. In addition, for the conclusion of "The activation of complement system was associated with the clinical manifestations, kidney pathological damage, and renal outcomes", I do not find enough data to support the activation of complement system was associated with the clinical manifestations and kidney pathological damage. 

Answer: Thank you for your comments. We have revised our conclusion in the abstract.

I have following comments which should be considered.

1. The definition of complete remission was stricter than other studies which was that proteinuria ≤ 0.3 g/24h with stable serum creatinine and serum albumin great than 35 g/L. Why the authors define complete remission as proteinuria less than or equal to 0.15 g/24h with stable serum creatinine?

Answer: Thank you for your comments. We are sorry for the slip of the pen. The definition of complete remission in our study was also proteinuria ≤ 0.3 g/24h with stable serum creatinine and serum albumin > 35 g/L. We corrected in the revised manuscript (page 6, paragraph 3, line 12; page 7, paragraph 1, line 1-2). 

2. The endpoint of renal outcome is end stage of renal disease, please clarify the definition.

Answer: In the study, ESRD was defined as eGFR < 15 ml/min/1.73m2 or initiation of renal replacement therapy. It has been added in the revised manuscript (page 7, paragraph 1, line 6-7).

3. I would suggest move the sentence of "The pathological variants of the 70 patients were defined according to the Columbia classification [2], which included 23 tip variant, 23 NOS variant, 21 cellular variant, 1 collapsing variant and 2 advanced FSGS (defined as the percentage of sclerosis/total glomeruli over 75%)" to renal histopathology section to make easy understanding.

Answer: Thank you. We have moved this sentence accordingly (page 7, paragraph 3, line 1-4).

4. A total of 70 patients were included in this study, but in Table 1 only 57 patients were evaluated the response to treatment, please clarify the reason of not evaluation 70 patients. The authors mentioned treatment in the method section but did not include them in Table 1. It would helpful to include them in Table 1.

Answer: In this study, medical data were collected from 70 patients at the time of kidney biopsy, while 57 of them had follow-up data. we discussed this limitation in the revised manuscript (page 28, paragraph 1, line 3-4). We analyzed the demographic and clinical parameters between patients with and without follow-up data in the following table. The results showed there was no significant difference between them, thus we thought it would not affect the results significantly. The treatments of all the patients have been added in Table 1 in the revised manuscript according to your suggestion （page 12）.

Table. The comparison of demographic and clinical parameters between patients with and without follow-up data.

 Patients with follow-up Patients without follow-up P value

Gender (male/female) 8/5 38/19 0.753

Age (years) (median, IQR) 28(18,49) 28(19,45) 0.821

Urine protein (g/24h)

(median, IQR) 7.45(5.14,11.50) 8.65(5.13,20.52) 0.272

Albumin (g/L) (median, IQR) 20.1(16.45,23.2) 18.0(15.8,27.65) 0.821

Serum creatinine (μmol/l)

(median, IQR) 83.5(62.30,148.25) 96.0(70.5,129.1) 0.504

eGFR(ml/min/1.73m2)

(median, IQR) 95.67(51.27,151.39) 80.45(48.47,119.71) 0.528

5. For Cox regression, did the authors included urinary Bb, C3a and C5a in the same model? Urinary Bb was correlated with C3a and C5a, therefore, the issue of collinearity will produce when all of them were included in the same model.

Answer: Thank you. Multicollinearity was tested using the variance inflation factor (VIF) method, with a VIF ≥ 5 indicating the presence of multicollinearity. Our results showed the VIF value of urinary Bb, C3a and C5a were 2.125, 2.401 and 1.759, respectively. So they did not have multicollinearity and urinary Bb, C3a and C5a were included in the same model (page 10, paragraph 1, line 6-7; page 22, paragraph 1, line 4-6).

---

## [Decision Letter · Decision Letter 1]

5 Jun 2020

Complement activation profile of patients with primary focal segmental glomerulosclerosis

PONE-D-20-06891R1

Dear Dr. huang,

We’re pleased to inform you that your manuscript has been judged scientifically suitable for publication and will be formally accepted for publication once it meets all outstanding technical requirements.

Kind regards,

Zhanjun Jia

Academic Editor

PLOS ONE

Additional Editor Comments (optional):

Reviewers' comments:

Reviewer's Responses to Questions

**Comments to the Author**

1. If the authors have adequately addressed your comments raised in a previous round of review and you feel that this manuscript is now acceptable for publication, you may indicate that here to bypass the “Comments to the Author” section, enter your conflict of interest statement in the “Confidential to Editor” section, and submit your "Accept" recommendation.

Reviewer #1: All comments have been addressed

Reviewer #2: All comments have been addressed

2. Is the manuscript technically sound, and do the data support the conclusions?

Reviewer #1: Yes

Reviewer #2: Yes

3. Has the statistical analysis been performed appropriately and rigorously? 

Reviewer #1: Yes

Reviewer #2: Yes

4. Have the authors made all data underlying the findings in their manuscript fully available?

Reviewer #1: Yes

Reviewer #2: Yes

5. Is the manuscript presented in an intelligible fashion and written in standard English?

Reviewer #1: Yes

Reviewer #2: Yes

6. Review Comments to the Author

Reviewer #1: (No Response)

Reviewer #2: The authors have addressed my questions. Thanks the authors considered my comments and revised the manuscript. I don't have additional question.

7. PLOS authors have the option to publish the peer review history of their article (what does this mean?). If published, this will include your full peer review and any attached files.

Reviewer #1: Yes: Ruochen Che

Reviewer #2: No

---

## [Editor Report · Acceptance letter]

11 Jun 2020

PONE-D-20-06891R1 

Complement activation profile of patients with primary focal segmental glomerulosclerosis 

Dear Dr. Huang:

I'm pleased to inform you that your manuscript has been deemed suitable for publication in PLOS ONE. Congratulations! Your manuscript is now with our production department. 

Kind regards, 

on behalf of

Dr. Zhanjun Jia 

Academic Editor

PLOS ONE